# Real-Time Monitoring of the Atrazine Degradation by Liquid Chromatography and High-Resolution Mass Spectrometry: Effect of Fenton Process and Ultrasound Treatment

**DOI:** 10.3390/molecules27249021

**Published:** 2022-12-17

**Authors:** Junting Hong, Nadia Boussetta, Gérald Enderlin, Nabil Grimi, Franck Merlier

**Affiliations:** 1Université de Technologie de Compiègne, UPJV, CNRS, Enzyme and Cell Engineering, Centre de Recherche Royallieu, CEDEX CS 60319, 60203 Compiègne, France; 2Université de Technologie de Compiègne, ESCOM, TIMR (Integrated Transformations of Renewable Matter), Centre de Recherche Royallieu, CEDEX CS 60319, 60203 Compiègne, France

**Keywords:** atrazine metabolite, online sampling-LC-HRMS, Fenton, ultrasound degradation

## Abstract

High resolution mass spectrometry (HRMS) was coupled with ultra-high-performance liquid chromatography (uHPLC) to monitor atrazine (ATZ) degradation process of Fenton/ultrasound (US) treatment in real time. Samples were automatically taken through a peristaltic pump, and then analysed by HPLC-HRMS. The injection in the mass spectrometer was performed every 4 min for 2 h. ATZ and its degradation metabolites were sampled and identified. Online Fenton experiments in different equivalents of Fenton reagents, online US experiments with/without *Fe*^2+^ and offline Fenton experiments were conducted. Higher equivalents of Fenton reagents promoted the degradation rate of ATZ and the generation of the late-products such as Ammeline (AM). Besides, adding *Fe*^2+^ accelerated ATZ degradation in US treatment. In offline Fenton, the degradation rate of ATZ was higher than that of online Fenton, suggesting the offline samples were still reacting in the vial. The online analysis precisely controls the effect of reagents over time through automatic sampling and rapid detection, which greatly improves the measurement accuracy. The experimental set up proposed here both prevents the degradation of potentially unstable metabolites and provides a good way to track each metabolite.

## 1. Introduction

Although the quantity of pesticides in water was significantly reduced in France between 2008 and 2018 [1], certain molecules, such as atrazine, banned in Europe since 2003, persist in the environment. Atrazine (2-chloro-4-ethylamino-6-isopropylamino-1,3,5-triazine) is a triazine herbicide with a wide range of application, for grassy and broadleaf weed control in corn, sugarcane, sorghum and other crops [2,3,4,5]. However, atrazine is also considered as one of the most toxic herbicides [6], because it can act as an endocrine disruptor that can produce damage to the endocrine system, causing a series of pathological changes and reproductive abnormalities [7,8]. In addition, atrazine is also a potential carcinogen due to negative impact on human health such as tumors, breast, ovarian, and uterine cancers as well as leukemia and lymphoma [9,10].

The literature reports numerous attempts aimed at degrading this water pesticide. The reported treatments are microwave-assisted photo reactions [11], high voltage electrical discharges [12], ultrasound [13,14], advanced oxidation processes (AOPs) [15,16] and bioremediation [17]. However, these treatments produce new molecules which are sometimes even more toxic to humans and the environment [15,18]. In Fenton oxidation, hydrogen peroxide (*H*_2_*O*_2_) is activated by ferrous (*Fe*^2+^) ions to generate hydroxyl radicals (*HO·*). In ultrasound treatment, high energy leads to water splitting, generating hydroxyl radicals (*HO·*) and hydrogen radicals (*H·*). Hydroxyl radicals (*HO·*) dominate the degradation of atrazine in Fenton oxidation and in ultrasound treatment [13].

In order to evaluate the performance of these treatments on pesticides degradation in water, it is essential to use analytical techniques to follow the evolution of its metabolites during the treatments. Most of the studies use chromatographic techniques with UV (LC-UV) [12,15,19] detection with or without mass spectrometry coupling (LC-MS). Compared with low-resolution tandem mass spectrometry (LC-MS/MS) [20], high-resolution mass spectrometry (LC-HRMS) [21] has higher accuracy in terms of metabolite characterization and exact mass specificity. In most cases, the required sampling for the kinetic monitoring of the degradation is carried out by spot sampling prior to transfer or storage [14,15,22]. When monitoring the kinetics of Fenton reaction, the presence of reagents in the medium can lead to errors in the estimation of metabolite amounts due to potential evolution of the samples and incomplete cessation of the reaction. To ensure reliable observation of the kinetics of each metabolite, automatic sampling can be combined with the LC-MS system [23]. Fenton reactions or ultrasound (US) treatments were previously monitored online with detections by Fourier transform infrared spectroscopy (FTIR) [24], illumination-assisted droplet spray ionization mass spectrometry IA-DSI-MS [25] or fluorescence [26].

The aim of the present study is to demonstrate the feasibility of monitoring the kinetics of a Fenton reaction or US treatment by means of LC-HRMS. This coupling will thus make it possible to increase both the sampling frequency and to minimize the sample analysis time, allowing access to the most unstable metabolites [25].

## 2. Results and Discussion

### 2.1. Influence of Sample Storage after High Frequency Ultrasound (HFUS) Treatment

Figure 1 shows the effect of storage time on the concentration of residual ATZ. At 0-day, the initial ATZ solutions were treated by US experiments with and without *Fe*^2+^ respectively (US+*Fe*^2+^ and US) at room temperature. Samples were taken after 2 h for LC-HRMS analysis, and then transferred to different temperatures refrigerators (4 °C, −20 °C, −80 °C) for analysis after 7 and 30 days.

High frequency ultrasound (HFUS) used at 525 kHz generates high temperature and a high-pressure region of bubbles in which hydroxyl radicals (*HO·*) and hydrogen radicals (*H·*) are produced by water splitting, and then the self-combination of radicals *HO·* gives hydrogen peroxides (Equations (1) and (2)) [27]. *Fe*^2+^ promotes the regeneration of radical (*HO·*) from hydrogen peroxide (Equation (3)) [28]. The combination of *Fe*^2+^ and HFUS acts in concert towards the production of hydroxyl radicals (*HO·*), which promotes the degradation of ATZ. As shown in Figure 1, at 0-day, the concentration of residual ATZ of US+*Fe*^2+^ treatment for 2 h was much lower than that of US treatment for 2 h.
(1)$$H_{2}O\;\;\left[+US\right]\;\to HO\cdot +H\cdot$$
(2)$$HO\cdot \;+\;HO\cdot \;\;\to \;\;\;H_{2}O_{2}$$
(3)$$Fe^{2+}\;+\;\;H_{2}O_{2}\;\;\to \;\;\;Fe^{3+}\;+\;HO\cdot \;\;+\;\;OH^{-}$$

In addition, with the increase of storage time, the concentration of residual ATZ decreased. Lower storage temperature prevented the reduction of residual ATZ to some extent, but still could not stop the further reactions during storage. This observation shows the interest of minimizing the interval time between treatment and its characterization in order to be able to monitor the least stable metabolites as well as possible.

### 2.2. Sonolysis and Fenton Reaction Reactor Coupled to LC-HRMS

To minimize degradation caused by the presence of ions or reactive molecules, automatic sampling was performed directly from the reactor containing the atrazine solution. Samples were then chromatographically separated. Conventional sampling followed by LC-MS or LC-MS/MS [13,29] analysis requires incompressible time of minutes or even days when the instruments are unavailable for analysis. Our experimental set up (see Section 3.5.6) makes it possible to take samples in real time and to analyze samples with a cycle of 4 min. A liquid circulation system was set up through flexible tubes, replaced regularly to limit adsorption and release phenomena, and a glass tip in suction was equipped for sampling. A specially designed peristaltic pump was used, so as to ensure flow rates on the order of 1 mL/min and limit the dead volume of the sampling loop. A high-pressure six-way valve ensured a 4 min cycle of sampling and injection into the HPLC system. This valve was switched to the analysis position, while an internal standard consisting of ATZ-D5 was injected via the HPLC syringe, and switched back to sampling mode 50 s after initial analysis time (T_0_), thus allowing a sampling cycle of 3.10 min before the next chromatographic separation.

Since the treatments last for several hours, it is important to add a constant quantity of internal standard to the sample, which normalizes the areas of metabolites’ peaks. Any degradation of the chromatographic separation was monitored for possible contamination of the ionization source by electrospray and mass drift of the analyzer, especially when *Fe*^2+^ ions are present in Fenton treatments as well as in US+*Fe*^2+^ treatments. Here, the more portable low-frequency ultrasound (LFUS) device at 50 kHz was used instead of the unportable high-frequency ultrasound (HFUS) device at 525 kHz.

Acquisition without prior selection of precursors is possible using the high-resolution mass spectrometer. This enables non-targeted analysis and helps to understand the reaction mechanism.

### 2.3. Related Compounds

The initial substrate ATZ, internal standard ATZ-D5, and other detected metabolites are shown in Table 1. The “metabolite level” corresponds to the potential level of transformations from ATZ to metabolite according to the degradation pathways (see Appendix A).

### 2.4. Degradation Rate of Atrazine

#### 2.4.1. Effect of Fenton Reagents Equivalents

Since the Fenton reagents play an important role in Fenton oxidation, experiments in different Fenton reagent equivalents were conducted. Here, the equivalent of Fenton reagents is the ratio of [*H*_2_*O*_2_] or [*Fe*^2+^] to ATZ, where the molar concentrations of *H*_2_*O*_2_ or *Fe*^2+^ are kept the same.

As shown in Figure 2, under low equivalents of Fenton reagents at 1 eq. and 2 eq. (The abbreviation “eq.” means equivalent.), the degradation rates were very slow. Only 25.7% and 37.2% of ATZ were degraded after two hours. However, increasing the equivalent of Fenton reagents effectively promoted the degradation of ATZ. It was found that ATZ was rapidly degraded within 20 min under high equivalents of Fenton reagents at 5 eq. and 10 eq. Then, the degradation rates slowed down to 89.8% and 92.7%, respectively, in 2 h.

The promotion effect of Fenton reagents could result from the production of hydroxyl radicals (*HO·*). Hydrogen peroxide (*H*_2_*O*_2_) was activated by ferrous ion (*Fe*^2+^) to generate hydroxyl radicals (*HO·*). Radicals *HO·* were strong oxidants to degrade organic compounds *RH* (Equations (3) and (4)). In addition, *Fe*^2+^ can be regenerated by the reduction of *Fe*^3+^ with *H*_2_*O*_2_ according to Equation (5). The need to use a large amount of hydrogen peroxide argues for the exclusion of a radical chain mechanism. High equivalents of Fenton reagents were more favorable.
(4)$$RH+HO\cdot \;\to \;H_{2}O+R\cdot$$
(5)$$Fe^{3+}+H_{2}O_{2}\to \;Fe^{2+}+HO_{2}\cdot +H^{+}$$

#### 2.4.2. Effect of LFUS Treatment with/without *Fe*^2+^

For online low-frequency ultrasound (LFUS) treatment at 50 kHz (70 W), adding or not adding *Fe*^2+^ showed an obvious difference in ATZ degradation. As shown in Figure 3, at the first 20 min, the presence or absence of *Fe*^2+^ had little effect on ATZ degradation, but after 30 min, the degradation rate of ATZ was significantly increased by the activation of *Fe*^2+^. A large number of transformations are in competition [30], according to the following admitted Equations (1)–(9). However, experimentally we can observe a beneficial effect on the generation of radicals both by ultrasound and the presence of ferrous ions.
(6)$$Fe^{2+}+HO\cdot \;\;\to \;\;Fe^{3+}+HO^{-}$$
(7)$$Fe^{2+}+HO_{2}\cdot \;\;\to \;\;Fe^{3+}+HO_{2}^{-}$$
(8)$$Fe^{3+}+HO_{2}\cdot \;\;\to \;\;Fe^{2+}+O_{2}+H^{+}$$
(9)$$H_{2}O_{2}\;\;\left[+US\right]\;\to \;\;2HO\cdot$$

Indeed, the mechanism of ultrasound is the implosion of the bubble with high energy, followed by the generation of hydroxyl radicals (*HO·*) and hydrogen radicals (*H·*) from water and hydrogen peroxide dissociation (Equations (1) and (9)). At the same time, *HO·* could be consumed by the in-situ *H*_2_*O*_2_ formation occurring in the bulk solution due to the radicals’ recombination (Equation (2)), which is not conducive to ATZ degradation. *Fe*^2+^ promotes ATZ degradation, possibly because it reacts with *H*_2_*O*_2_ and re-releases *HO·* (Equation (3)). Other transformations and recombination involving the radicals are possible and are not exhaustively reported.

#### 2.4.3. Atrazine Degradation of Offline Fenton Experiments

In order to study the effect of the Fenton oxidation reaction time on ATZ degradation, offline Fenton experiments were conducted. As shown in Figure 4, the reaction time had a great influence on the ATZ degradation. Regardless of the equivalents of Fenton reagents, the degradation rate of ATZ increased with time, and it was almost completely degraded after 8 h. In addition, comparing with Figure 2, at 2 h, for the same equivalent of Fenton reagents, the ATZ degradation rates of offline Fenton seem to be greater than that of online Fenton. For example, after 2 h, the ATZ degradation rates of online Fenton at 2 eq., 5 eq. and 10 eq. are 37%, 90% and 93%, respectively (Figure 2), while that of offline Fenton at 2 eq., 5 eq. and 10 eq. are 47%, 90% and 94% (Figure 4). This is probably because the solution in the vial was still reacting after sampling, as we were able to observe during the stability tests (Figure 1). So, in order to improve accuracy, online HPLC-HRMS analysis is necessary, which can monitor the reaction process in real time.

### 2.5. Kinetics of Metabolites

The equivalents of Fenton reagents affect the production of metabolites during the reaction. As shown in Figure 5, the kinetics of metabolites varies from the different equivalents of Fenton reagents. Generally, the kinetics of metabolites in high equivalents of the Fenton reagents system are more complicated, with more products, especially late degradation products such as AM [13]. With 1 and 2 eq. of oxidants, corresponding to the Fenton process (Figure 5a,b), 8 metabolites (DEA, DIA, CDET, CDIT, ODIT, CNIT, HAHT, and CVIT) were detected, but their changing trends were different. With 1 eq. Fenton reagents (Figure 5a), CDIT increased rapidly in the first 30 min until it reached about 4% and tended to balance; HAHT was more than CDIT in the first 20 min, but then its growth rate slowed down until the final 3.4%; other main products were DIA, CVIT and ODIT. With 2 eq. Fenton reagents (Figure 5b), CDIT reached balance earlier and was greater than 4%; HAHT increased faster and eventually surpassed CDIT; other main products were DIA, CVIT and CNIT.

With 5 eq. and 10 eq. Fenton reagents (Figure 5c,d), except the above 8 metabolites (DEA, DIA, CDET, CDIT, ODIT, CNIT, HAHT, and CVIT), another 5 metabolites (DDA, CDAT, CDDT, ODDT, and AM) were detected. In addition, a ring-broken compound CBOI was found in the 10 eq. Fenton process. In general, for a high equivalents Fenton process, the metabolites increased rapidly in the first 20 min due to the quick activation of *H*_2_*O*_2_ by *Fe*^2+^, and also more late degradation products were generated due to a higher concentration of hydroxyl radicals (*HO·*). With 5 eq. Fenton reagents (Figure 5c), CDIT, HAHT, DIA, CVIT, and CNIT were still the main products; CDET increased in the first 30 min and then decreased; other main products were DEA, ODDT, and CDDT, which increased slowly; smaller molecules DDA and AM appeared. In the 10 eq. Fenton process (Figure 5d), the main products CDDT, CDIT, DIA, and DEA were all firstly increased and then decreased; other main products CDAT, CDET, ODIT, ODDT, and HAHT were all increased until they reached balance; DDA had a small peak around 55 min, while AM was still a small amount as well as the ring-broken compound CBOI.

As shown in Figure 6, ferrous ion *Fe*^2+^ has an effect on the kinetics of metabolites during the US process. Six metabolites (DEA, DIA, CDIT, ODIT, and HA) were detected in US treatment without *Fe*^2+^ (Figure 6a), while 3 more metabolites (DEHA, CDET, and ODET) were detected in US treatment with *Fe*^2+^ (Figure 6b). When using ultrasound without *Fe*^2+^, the metabolites increased very slowly in the first 20 min. This is probably because the initial US cavitation had not produced enough hydroxyl radicals (*HO·*). The main product DIA increased to the maximum at 40 min and then decreased, while the second main product CDIT increased in the first 60 min and then reached balance. DEA increased slowly all the time; ODIT increased suddenly at 40 min and then approached DEA. When using ultrasound with *Fe*^2+^, DIA and DEA increased to the maximum at 30 min and 76 min, respectively, and then decreased. ODIT linearly increased. In comparison, adding *Fe*^2+^ promoted the generation of dealkylation products (DIA and DEA), and the dichlorination products (ODIT, ODET, HA and DEHA), but inhibited the generation of acylation products CDIT.

In Fenton oxidation, the substitution of chlorine by hydroxyl is favored by the increase of Fenton reagents’ amount (Table 2). However, the ratio of metabolites without chlorine to metabolites with chlorine is relatively stable over time in the case of the Fenton reaction, regardless of the amount of Fenton reagents once the threshold of 50 min has been reached. This result is consistent with the previous work [31] which proposed a practical model for predicting ATZ decay performance based on the Fenton reagents. In this model, Fenton’s process can be characterized as a process with two stages (i.e., a rapid stage I followed by a retarded stage II). In rapid stage I, the rapidly generated *HO·* radicals prioritized the allylic-oxidation and the dealkylation of ATZ, competing with the dechlorination of ATZ. In retarded stage II, *HO·* radicals were deficient, but some less reactive radicals (*HOO·* and *O_2_·*) presented. These less reactive radicals were not active enough to oxidize ATZ, but were still useful in oxidizing selective intermediates. Therefore, although dechlorinated products were slowly increasing, chlorinated products remained at a high level throughout the Fenton process due to the rapid initial production and consumption of *HO·* radicals.

As proposed in previous works [13,32], in the US system, ATZ molecules mainly congregate and decay at the gas bubble interfaces, and only a small proportion decayed in the bulk solution because *HO·* radicals concentrate and react at the surface of bubbles. Radicals *HO·* tend to degrade ATZ through allylic oxidation and dealkylation. So, at the bubble interfaces, ATZ is rapidly converted to chlorinated products. These chlorinated products and ATZ re-diffused to the bubble interface will be continuously degraded by newly generated *HO·* radicals, followed by the gradual increase of dechlorinated products. Furthermore, *Fe*^2+^ can release *C* radicals through the Fenton reaction with *H*_2_*O*_2_ generated from the self-combination of *HO·* radicals. Therefore, in the US+*Fe*^2+^ system, there are more *HO·* radicals than in the US-only system, which contributes to the increased ratio of dechlorinated products to chlorinated products (Table 2). This ratio increases obviously over time, in contrast to Fenton’s, because *HO·* radicals are continuously generated in the US system but rapidly generated and depleted in the Fenton system.

For Fenton oxidation, whatever the amount of iron, amidation is the most observed process followed by dealkylation after 50 min (Table 3), which is in agreement with the most recent work [29]. In the context of the use of ultrasound, a simple transformation of ATZ to HA is observed, as also observed by Shi when using a catalyst [33]. Fenton reactions mainly produce metabolites by amildation. Petrier [14] proposed a mechanism suited to the operating conditions. Ultrasonic reactions produce more abundant dealkylated compounds, to the detriment of amides, hydroxy and dehydrogenated.

Preferential oxidation paths are observed depending on the quantity of hydroxyl radicals produced and available. The greater this quantity of radicals, the deeper the oxidation, up to an opening of the aromatic ring. In the light of these first results, it would be interesting to follow some degradation products such as CNIT to identify their evolution either towards CDIT or towards imines that can serve as intermediaries for the creation of dealkylated metabolites.

## 3. Materials and Methods

### 3.1. Chemicals and Reagents

Solvents and formic acid were purchased from Biosolve Chimie with UPLC-MS grade (Dieuze, Moselle, France). Atrazine (ATZ), atrazine-D5 (ATZ-D5), deisopropylatrazine (DIA), deethylatrazine (DEA), deethyldeisopropylatrazine (DEDIA), hydroxyatrazine (HA) and ammeline (AM) were purchased from Sigma-Aldrich (St. Quentin Fallavier, France). Buffers were prepared with Milli-Q water, purified using a Milli-Q system (Millipore, Molsheim, France).

### 3.2. Internal Standard (IS) Preparation

Atrazine isotopically labelled with 5 deuterium (ATZ-D5) was used as an internal standard. To prepare a working IS solution, ATZ-D5 solution of 2 mg/L in acetonitrile was diluted to 15 ng/mL in distilled water.

### 3.3. Preparation of Calibration Standard and Quality Control (QC) Samples

Stock solutions of ATZ, DEA, DEDIA, AM, HA and DIA were prepared separately at 50 mg/L by dissolving accurately-weighed amounts of the compound in methanol. The working standard solutions of ATZ were prepared by serial dilution of the ATZ stock solutions with distilled water, to give final concentrations of 10, 5, 2, 1, 0.5 and 0.25 ng/mL. Calibration curve standard samples of ATZ were prepared by spiking 20 µL of IS solution with 180 µL of working standard solutions. QC samples were prepared in the same way.

### 3.4. Sample Preparation

An automatic sampling method was used for the online analysis. Online samples were taken every 4 min. The reaction solution (4 µL) was automatically injected into the HPLC, while the IS solution (2 µL) was added when switching the sampling valve by HPLC auto injector. For offline analysis, the reaction solution (50 µL) was manually mixed with the IS solution (20 µL). Then the 1 µL mixed solution was injected into the LC-HRMS (see Section 3.5.5).

### 3.5. Instrumentation and Analytical Conditions

#### 3.5.1. Offline HFUS Experiment with/without Fe^2+^

Two initial solutions of ATZ (20 mg/L, 50 mL, 4.64 µmol) with/without 10 eq. *Fe*^2+^ (13 mg, 46.4 µmol *FeSO_4_∙7H_2_O*) were treated by ultrasound (525 kHz, 80 W) for 2 h. After these two experiments, 2 × 9 samples were taken (1 mL of final treatment solution was added to 1.5 mL of vial), 2 × 3 of which were used for 0-day analysis, and the rest were stored in different refrigerators (4 °C, −20 °C, and −80 °C). Absolute quantification was performed by LC-UV at 260 nm without addition of internal standard.

#### 3.5.2. Online Fenton Experiment

The initial solution of ATZ (20 mg/L, 92.72 µmol/L) was prepared by dissolving ATZ (5 mg, 23.18 µmol, 215.68 g/mol) into 250 mL of distilled water. Different equivalents of Fenton reagents 10 eq., 5 eq., 2 eq. and 1 eq. were added, respectively, to four initial solutions of ATZ (15 mL, 1.392 µmol): *FeSO_4_∙7H_2_O* (10 eq., 3.9 mg, 13.92 µmol) and 30% *H*_2_*O*_2_ solution (10 eq., 1.422 mL, 13.92 µmol); *FeSO_4_∙7H_2_O* (5 eq., 1.95 mg, 6.96 µmol) and 30% *H*_2_*O*_2_ solution (5 eq., 0.711 mL, 6.96 µmol); *FeSO_4_∙7H_2_O* (2 eq., 1.56 mg, 2.784 µmol) and 30% *H*_2_*O*_2_ solution (2 eq., 0.284 mL, 2.784 µmol); *FeSO_4_∙7H_2_O* (1 eq., 0.39 mg, 1.392 µmol) and 30% *H*_2_*O*_2_ solution (1 eq., 0.142 mL, 1.392 µmol). Those four reactions were incubated for 2 h with a magnetic stirrer at 500 rpm. The time t = 0 min corresponds to the first LCMS sample tested. Fenton reagents were added at t = 4 min.

#### 3.5.3. Online LFUS Experiment with/without Fe^2+^

The initial solution of ATZ (15 mL, 1.392 µmol) without *Fe*^2+^ was treated by ultrasound (50 kHz, 70 W) for 2 h. Then, in another controlled experiment with *Fe*^2+^, the initial solution of ATZ (15 mL, 1.392 µmol) was treated by ultrasound (50 kHz, 70 W) in the same way, but after 4 min *FeSO_4_∙7H_2_O* (10 eq., 3.9 mg, 13.92 µmol) was added. As the temperature rose during ultrasonic treatment, a water circulating cooling device was used on the outer wall of the reaction system.

#### 3.5.4. Offline Fenton Experiment

Nine initial atrazine solutions (7.5 mL, 0.696 µmol) were divided into three groups, and then different equivalents of Fenton reagents 10 eq., 5 eq. and 2 eq. were added, respectively. The reactions were incubated on a magnetic stirrer at 500 rpm. Samples were taken at 0 h, 1 h, 2 h, 4 h, 8 h and 22 h. At time t = 0 min, Fenton reagents were not added. At time t = 4 min, Fenton reagents were added.

#### 3.5.5. HPLC-HRMS

Atrazine and metabolites detection and semi quantitative evaluation were performed by LC-HRMS. The HPLC system (Infinity 1290, Agilent Technologies, France) with DAD, was connected to a Q-TOF micro hybrid quadrupole time of flight mass spectrometer (Agilent 6538, Agilent Technologies, France) with electrospray ionization (ESI). HPLC was carried out on a Thermo Hypersyl Gold C18 (USP L1) column (100 × 2.1 mm, 1.9 µm, 175 A), connected to an Agilent Infinity 1290 HPLC at 40 °C. The solvent system was A: 0.1% formic acid in H_2_O and B: Acetonitrile. The gradient program began with 5% B, held at % for 0.3 min and ramped to 30% B at 1.7 min and to 95% at 3.5 min, held at 95% for 0.5 min, until decreased to initial condition and held at 5 % for 0.5 min. The flow rate was set at 0.600 mL/min. All compounds’ responses were measured in ESI+ and were calibrated externally. The ESI Gas Temp was 350 °C, at electrospray voltage +3800 V. Drying Gaz was set at 10 L/min and Nebuliser was at 30 psi. Fragment voltage was set at 110 V. HRMS spectrum was registered at 5 Hz in the mass range of 50 to 1200 *m*/*z* with internal calibration.

#### 3.5.6. HPLC-HRMS Online Setup

The system consists of a 20 mL glass reactor with a double wall for temperature control and an ultrasound probe (Figure 7a). Sampling is carried out by suction using a peristaltic pump (Welco WMP1-F1.6FB-WP), equipped with a brushless motor. This pump was used with a voltage of 13 V DC corresponding to a flow rate of 1 mL/min. A glass tip was inserted into a flexible tube with 1.6 mm internal diameter connected to a 6-way high pressure valve (Agilent 1316C module). A sampling loop consists of a capillary tube type thermo viper SST “Black” (0.18 mm ID × 150 mm). In the initial position, the mixture is recycled to the reactor. At the time of analysis, the content of the loop is sent to the automatic injector which simultaneously injects 1 µL of an ATZ-D5 (IS) solution before being separated and analysed on the LC-ESI-HRMS system. After 50 s, the valve switches back to its initial state in order to prepare the next sample. The sampling and analysis cycle have a 4 min duration.

### 3.6. Data Analysis

Software MassHunter (Version B.07.00, Agilent Technologies, Santa Clara, CA 95051, USA), was used for data processing. For Mass Profiler Pro workflow, an untargeted compound was generated by MFE algorithm.

### 3.7. Compound Identification

The main metabolites of atrazine were validated by the conjunction of exact mass, MS/MS fragmentations, and retention time from standards [34]. A complementary list of metabolites was established from the literature and sought from the exact mass of the mono-isotope ion, if necessary, verified by MS/MS and compared in relative retention time compared to data from the literature (Appendix A).

## 4. Conclusions

In this work, the atrazine degradation process of Fenton/US treatment was monitored in real-time by an online HPLC-HRMS analysis system. Compared with offline analysis, online analysis can avoid additional reactions after sampling, which greatly improved measurement accuracy. In addition, this online method is effective because of automatic sampling and it only takes 4 min analysis time. During analysis, ATZ, ATZ-D5, and seventeen metabolites were identified by accurate mass measurement, which provided abundant information on the atrazine degradation process. The results showed that high equivalents of Fenton reagents promoted the degradation rate of ATZ and the generation of late degradation products such as AM. In addition, adding *Fe*^2+^ accelerated ATZ degradation in US treatment. The kinetics of metabolites in different conditions was useful for mechanisms research. However, because of the complexity of the reaction process, there are still compounds that are difficult to detect, especially the small and trace molecules. Therefore, for follow-up work, set-up improvement and conditions optimization will be conducted, so as to expand the upper and lower detection limits and improve detection accuracy.

## Figures and Tables

**Figure 1 molecules-27-09021-f001:**
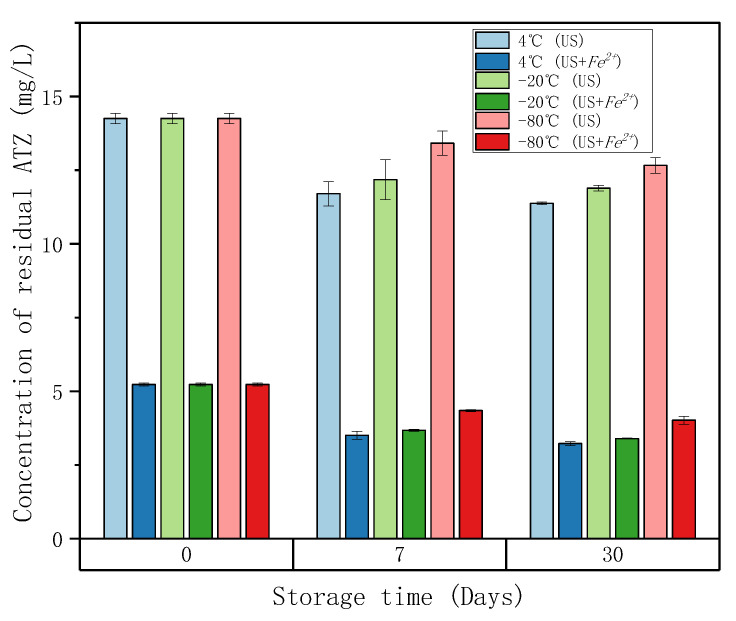
The effect of storage time on the concentration of residual ATZ. At 0-day, the initial ATZ solutions were treated by HFUS without or with *Fe*^2+^, US or US+*Fe*^2+^. Conditions: [ATZ]_0_ = 0.093 mmol/L; [*FeSO*_4_·7*H*_2_*O*]_0_ = 0.93 mmol/L for treatment US+*Fe*^2+^; volume = 50 mL; ultrasound frequency = 525 kHz; reaction time = 2 h (see Section 3.5.1). Samples were taken and then stored at different temperatures (4 °C, −20 °C, and −80 °C).

**Figure 2 molecules-27-09021-f002:**
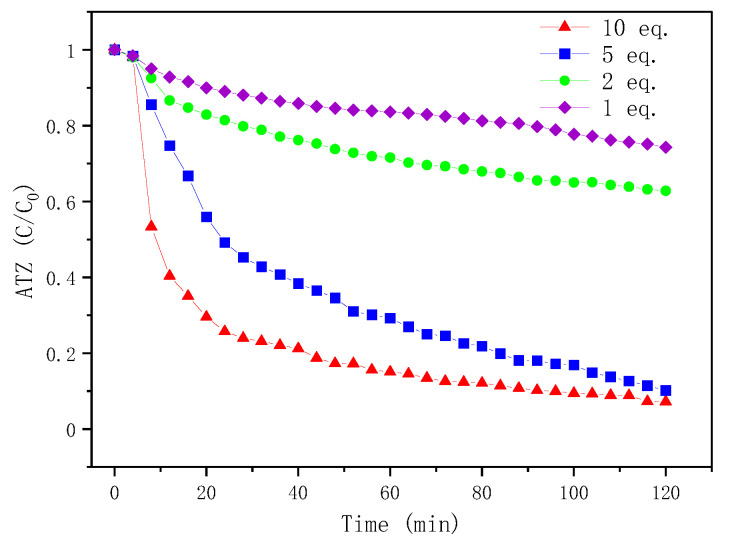
Time profiles of ATZ degradation for different Fenton reagent equivalents. Conditions: [ATZ]_0_ = 0.093 mmol/L; [*FeSO_4_·7H_2_O*]_0_ = [*H*_2_*O*_2_]_0_ = 0.93 mmol/L, 0.465 mmol/L, 0.186 mmol/L and 0.093 mmol/L for 10 eq., 5 eq., 2 eq. and 1 eq. Fenton reagents; volume = 15 mL; reaction time = 2 h (see Section 3.5.2).

**Figure 3 molecules-27-09021-f003:**
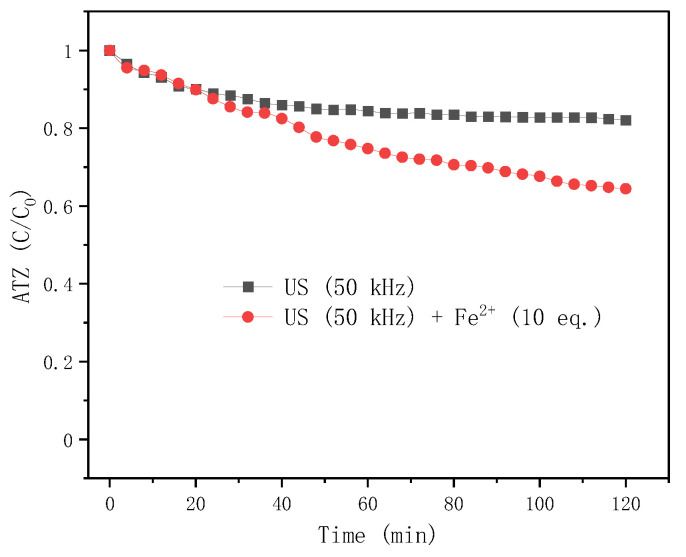
Time profiles of ATZ degradation during LFUS treatment with or without *Fe*^2+^ (US or US+ *Fe*^2+^). Conditions: [ATZ]_0_ = 0.093 mmol/L; [*FeSO*_4_·7*H*_2_*O*]_0_ = 0.93 mmol/L for treatment US+ *Fe*^2+^; volume = 15 mL; ultrasound frequency = 50 kHz; reaction time = 2 h (see Section 3.5.3).

**Figure 4 molecules-27-09021-f004:**
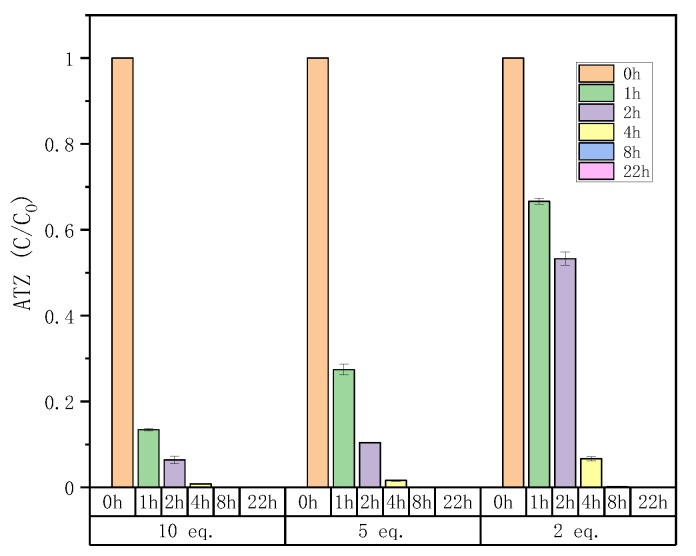
Time profiles of ATZ degradation during offline Fenton treatment for different equivalents of Fenton reagents. Conditions: [ATZ]_0_ = 0.093 mmol/L; [*FeSO_4_·7H_2_O*]_0_ = 0.93 mmol/L, 0.465 mmol/L and 0.186 mmol/L for 10 eq., 5 eq. and 2 eq.; volume = 15 mL; ultrasound frequency = 50 kHz (see Section 3.5.4).

**Figure 5 molecules-27-09021-f005:**
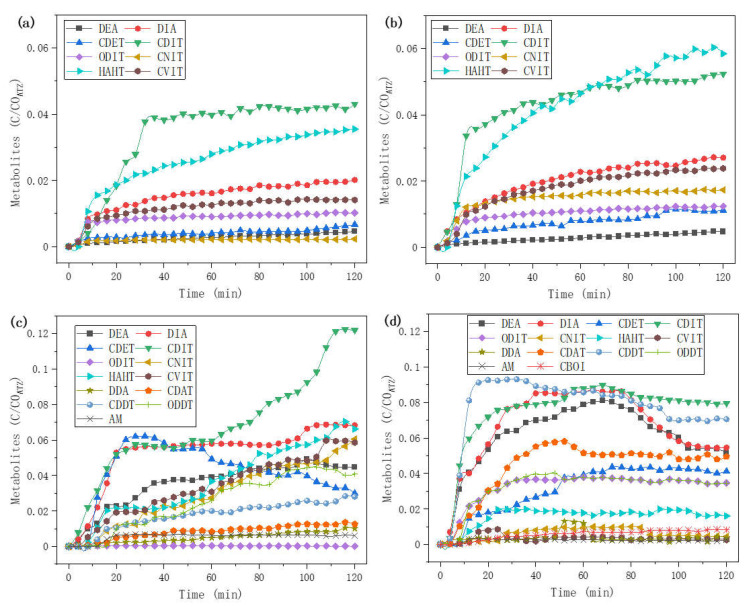
Kinetics of metabolites in online Fenton process. (**a**) Fenton reagents 1 eq.; (**b**) Fenton reagents 2 eq.; (**c**) Fenton reagents 5 eq.; (**d**) Fenton reagents 10 eq. (see Section 3.5.2).

**Figure 6 molecules-27-09021-f006:**
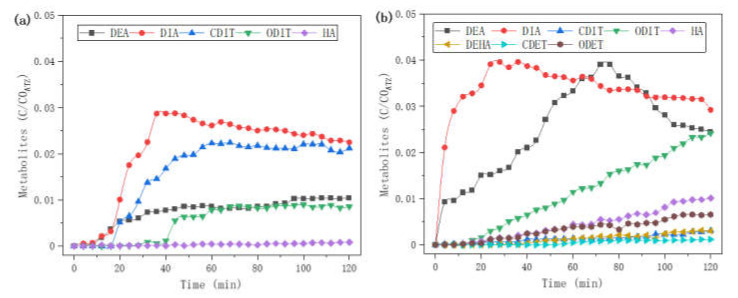
Kinetics of metabolites in online LFUS 50 kHz process. (**a**) Without 10 eq. *Fe*^2+^; (**b**) With 10 eq. *Fe*^2+^ (see Section 3.5.3).

**Figure 7 molecules-27-09021-f007:**
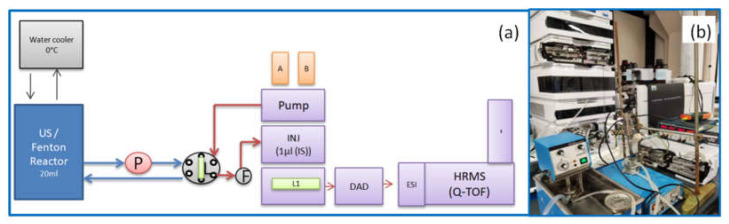
(**a**) Scheme of online sampling setup where, P is the peristaltic pump, F is the 0.2 µm filter, A and B is HPLC eluant, L1 is the C18 column. (**b**) Photo of actual system.

**Table 1 molecules-27-09021-t001:** Information of related compounds.

Entries	Chemical Structure	Molecular Formula	Abbreviation	Name	*m*/*z*	Retention Time (min)	MetaboliteLevel
Online	Offline
1	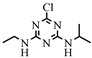	C_8_H_14_ClN_5_	ATZ	Atrazine	216.1010	2.558	2.178	0
2	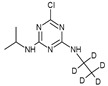	C_8_H_9_D_5_ClN_5_	ATZ-D5	Atrazine-D5	221.1324	2.579	2.168	\
3	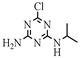	C_6_H_10_ClN_5_	DEA	Deethylatrazine	188.0698	1.910	1.751	1
4	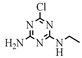	C_5_H_8_ClN_5_	DIA	Deisopropylatrazine	174.0541	1.620	1.463	1
5	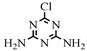	C_3_H_4_ClN_5_	DDA	Didealkylatrazine	146.0228	1.023	0.905	2
6	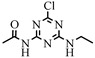	C_7_H_10_ClN_5_O	CDET	Simazine amide	216.0647	1.820	1.691	1
7	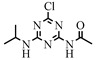	C_8_H_12_ClN_5_O	CDIT	Atrazine amide	230.0804	2.091	1.875	1
8	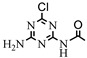	C_5_H_6_ClN_5_O	CDAT	Deisopropylatrazine amide	188.0334	1.264	1.257	2
9	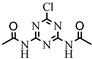	C_7_H_8_ClN_5_ O_2_	CDDT	N,N’-(6-Chloro-1,3,5-triazine-2,4-diyl)diacetamide	230.0440	1.451	1.407	3
10	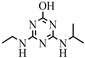	C_8_H_15_N_5_O	HA	Hydroxyatrazine	198.1350	1.233		1
11	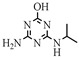	C_6_H_11_N_5_O	DEHA	Deethylhydroxyatrazine	170.1037	0.810		2
12	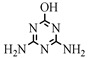	C_3_H_5_N_5_O	AM	Ammeline	128.0567	0.490	0.475	3
13	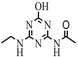	C_7_H_11_N_5_O_2_	ODET	N-[6-(Ethylamino)-4-oxo-1,4-dihydro-1,3,5-triazin-2-yl]acetamide	198.0986	1.082	1.048	2
14	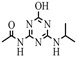	C_8_H_13_N_5_O_2_	ODIT	Hydroxyatrazine amide	212.1142	1.213	1.067	2
15	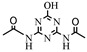	C_7_H_9_N_5_O_3_	ODDT	N,N’-(6-hydroxy-1,3,5-triazine-2,4-diyl)diacetamide	212.0778	1.141	1.013	3
16	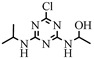	C_8_H_14_ClN_5_O	CNIT	1-({4-chloro-6-[(propan-2-yl)amino]-1,3,5-triazin-2-yl}amino)ethan-1-ol	232.0960	1.868	1.732	1
17	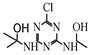	C_8_H_14_ClN_5_O_2_	HAHT	2-({4-chloro-6-[(1-hydroxyethyl)amino]-1,3,5-triazin-2-yl}amino)propan-2-ol	248.0909	2.212	1.933	2
18	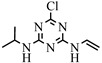	C_8_H_12_ClN_5_	CVIT	6-chloro-N2-ethenyl-N4-(propan-2-yl)-1,3,5-triazine-2,4-diamine	214.0854	2.070		2
19	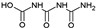	C_3_H_5_N_3_O_4_	CBOI	1-carboxybiuret	148.0353	0.416		4

**Table 2 molecules-27-09021-t002:** Distribution of metabolites by ring substituent during treatments after 50 and 100 min, respectively *.

Treatment	50 min				100 min			
	Cl	OH	Ring Opening	C/C_0_(ATZ)	Cl	OH	Ring Opening	C/C_0_(ATZ)
^a^ Fenton_1 eq.	92.56%	7.44%		84.08%	93.32%	6.68%		77.75%
^a^ Fenton_2 eq.	94.20%	5.80%		72.84%	94.45%	5.55%		65.00%
^a^ Fenton_5 eq.	91.23%	8.77%		31.02%	89.30%	10.70%		16.90%
^a^ Fenton_10 eq.	85.70%	13.19%	1.10%	17.27%	83.37%	14.92%	1.71%	9.46%
US_50 kHz	89.86%	10.14%		84.73%	86.44%	13.56%		82.81%
${}^{b}\;\mathrm{US}\text{\_}50\;\mathrm{kHz}+Fe^{2+}$	81.01%	18.99%		76.80%	64.54%	35.46%		67.62%

* The sum of the percentages containing chlorine, hydroxyl groups, and ring-opening products was 100%. ^a^ Keeping the same molar concentrations of *FeSO_4_·7H_2_O* and *H*_2_*O*_2_, different equivalents of Fenton reagents were used on the degradation of atrazine by Fenton oxidation. ^b^ 10 eq. *FeSO_4_·7H_2_O* was added to the degradation of atrazine by ultrasound at 50 kHz.

**Table 3 molecules-27-09021-t003:** Distributions of metabolites by type of transformation during treatments after 50 and 100 min, respectively *.

Treatment	Amidation	Dealkylation	Dehydrogenation	Hydroxylation	Substitution (Conversion of ATZ to HA)
50 min					
Fenton_1 eq.	47.65%	16.58%	10.81%	24.96%	
Fenton_2 eq.	38.33%	14.15%	11.37%	36.15%	
Fenton_5 eq.	47.46%	29.32%	8.45%	14.77%	
Fenton_10 eq.	61.77%	32.39%	0.70%	5.14%	
US_50 kHz	41.94%	57.51%			0.55%
$\mathrm{US}\text{\_}50\;\mathrm{kHz}+Fe^{2+}$	15.82%	80.14%			4.04%
100 min					
Fenton_1 eq.	43.96%	17.23%	11.01%	27.80%	
Fenton_2 eq.	37.40%	14.21%	11.56%	36.83%	
Fenton_5 eq.	43.58%	26.04%	9.64%	20.75%	
Fenton_10 eq.	67.20%	26.82%	0.71%	5.27%	
US_50 kHz	47.09%	52.04%			0.88%
$\mathrm{US}\text{\_}50\;\mathrm{kHz}+Fe^{2+}$	28.44%	63.32%			8.25%

* The sum of the percentages of each transformation was 100%.

## Data Availability

Not applicable.

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
