# Peer review of "Real-Time Monitoring of the Atrazine Degradation by Liquid Chromatography and High-Resolution Mass Spectrometry: Effect of Fenton Process and Ultrasound Treatment"

_molecules, 2022, doi:10.3390/molecules27249021_

Round 1

Reviewer 1 Report

This manuscript reports on the real-time product monitoring of ATZ degradation by Mass spectrometry in Fenton, ultrasound and ultrasound/Fe2+ systems. While the detailed mechanism may fall out of the scope of this study, the product data are interesting and can provide a guide for some future research. The following few comments can be considered.

Specific comments

1.       L21-23. The sentence is not complete. L34, “causing”, etc. Please limit such minor editorial and grammatical mistakes.

2.       L44-46. The authors tend to suggest that the degradation of atrazine in the US system is by hydroxyl radical. Is there any possibility for the pyrolysis to occur? In fact, at L246 and Table 2, different routes of product generation suggest that US+Fe may have a different reaction mechanism than the Fenton reaction, which based solely on the hydroxyl radical. Therefore, if no data are available, some literature information on the mechanism such as radical contribution can be added.

3.       L71-72 vs L83-86, L297 and Fig 1. There is some confusion. Are the experiments (namely for the 0-day samples) conducted at different temperatures, or are they done at room temperature before transferring to the different temperature refrigerators?

4.       L125-127, Table 1, and Fig SI.2. Not sure what it means by “metabolite level”. It seems to be the “steps” the product takes to generate. Compound CVIT is formed directly from ATZ in Fig SI.2, but it has a level of 2 in the table. Compound ODDT is formed from CDDT, but they have the same levels. Please recheck (these and maybe others).

5.       L141-142, L150-151, L294-295. The Fenton regent “equivalent” in this manuscript seems to be the ratio of [H2O2] or [Fe2+] to ATZ, where the molar concentrations of H2O2 and Fe2+ are kept the same. Please add the definition of the equivalent at the beginning of section 2.4.1.

6.       L259, Table 3. At 50 min, the sum is not 100%.

7.       L309. Delete “HPLC” if MS is the analytical tool.

8.       Fig SI.2. Route “ATZà CEID” does not make sense. Maybe eliminate CEID from results and discussion.  

Author Response

We thank you for your constructive comments. you will find attached our answers to your questions and remarks.

Reviewer 2 Report

thanks, very nice paper and a very good concept for the investigation of degradation mechanisms - especially the fast tracking of metabolites has a high potential for the investigation of mechanisms

Author Response

(The authors gave the same response as above.)

Reviewer 3 Report

The topic of the research is interesting and valuable. The authors should improved significantly the manuscript: it is quite difficult to follow what they try to show. The experiments are not clear enough, nor the schemes that they explain. It is not quite understandable, even when the topic is interesting.

The English style should be improved quite a lot; that would also help to follow the research.

Author Response

(The authors gave the same response as above.)

Reviewer 4 Report

The manuscript demonstrates the feasibility of monitoring the kinetics of a Fenton reaction or US treatment by means of LC-HRMS. The experimental design is correct and well executed, references are adequate and the conclusions are sufficiently supported by the data. In my opinion, the manuscript is suitable for publication in this journal.

Author Response

(The authors gave the same response as above.)

Round 2

Reviewer 3 Report

L. 48: the use of "even" is confusing

L. 52: plese specify the meaning of "the later" here

L. 68: results

L. 71: change the position of respectively

L. 151: "aading or not..."

L. 152: I do not understand what the authors mean saying "at the beginning"

L. 153: "...degradation, but..."

L. 162: please rephrase when saying "...degradation. And..."

L. 202: which 9 metabolites?

L. 205: increased

L. 210: smaller

L. 244-246: please rephrase this sentence starting with "In retarded stage..."

L. 269: were

L. 286: materials and methods

L. 351: decreased

L. 352: responses

L. 388: it only takes 4 min analysis time

L. 393: mechanisms

Author Response

We thank you for your constructive criticism, you will find attached the modifications made to the manuscript.

Best regards
